

# Weakened and Irregular Miocene Climate Response to Orbital Forcing compared to the modern day

Yurui Zhang[1*], Jilin Wei[2,3], Zhen Li[1], Nan Dai[1], Weipeng Zheng[2,3,4] , Qiuzhen Yin[5], Agatha M. de Boer[6], Zhengguo Shi[7,8], Lixia Zhang[2]

[1]State Key Laboratory of Marine Environmental Science, College of Ocean & Earth Sciences, Xiamen University, Xiamen, China

[2]State Key Laboratory of Earth System Numerical Modeling and Application, Institute of Atmospheric Physics, Chinese Academy of Sciences, Beijing, China

[3]College of Earth and Planetary Sciences, University of Chinese Academy of Sciences, Beijing, China

[4]Earth System Numerical Simulation Science Center, Institute of Atmospheric Physics, Chinese Academy of Sciences, Beijing, China

[5]Earth and Climate Research Center, Earth and Life Institute, Universit écatholique de Louvain, Louvain-la-Neuve, Belgium

[6]Department of Geological Sciences, Bolin Centre for Climate Research, Stockholm University, Sweden

[7]State Key Laboratory of Loess Science, Institute of Earth Environment, Chinese Academy of Sciences, Xi'an, China

[8]Institute of Global Environmental Change, Xi'an Jiaotong University, Xi'an, China

*Correspondence to*: Yurui Zhang (yuruizhang@xmu.edu.cn)

**Abstract.** Orbital forcing is a well-established driver of Pleistocene glacial-interglacial cycles, but its role in warmer climates remains less clear. Using climate model simulation, we assess temperature response to maximum and minimum boreal summer insolation during the Miocene and pre-industrial (PI) time. Both exhibit broadly anti-phased responses, but the Miocene shows weaker and less coherent patterns. Three notable differences emerge: (1) Boreal land regions respond less strongly in the Miocene due to dampened albedo feedbacks from altered vegetation; (2) Tropical Africa experiences stronger cooling under high insolation, driven by an intensified hydrological cycle with a broader Tethys Ocean under warm climate; (3) The Southern Ocean warms unexpectedly under low insolation, linked to sea ice involved albedo feedback. Lower internal temperature variability in the Miocene suggests enhanced climate stability and weaker orbital pacing. These findings highlight the importance of background climate state in shaping orbital-scale climate and interpreting proxy records.

## 1 Introduction

It has been widely accepted that orbital forcing was at the origin of the Pleistocene glacial-interglacial cycles. A strong evidence is the good correlation between the periodicities of  paleoclimate records and those of the astronomical parameters [Hays et al., 1976; Berger, 1978; *Lisiecki and Raymo*, 2005]. Earth's orbital parameters—eccentricity, obliquity, and precession—regulate the timing and intensity of climate variability by altering the



seasonal and spatial distribution of incoming solar radiation (Berger, 1978; Hays et al., 1976; Milanković, 1969).
Summer insolation in the high latitudes of the Northern Hemisphere (NH) has been suggested as a key driver to
control the glacial-interglacial cycle through climate feedbacks (Milanković, 1969). Elevated NH summer insolation
warms high-latitudes and enhances rainfall across regions from Africa to Southeast Asia by strengthening early-
summer land heating and shifting convection inland (Battisti et al., 2014; Bosmans et al., 2018; Dai et al., 2024;
Herold et al., 2012; Yin et al., 2012).
Despite being a key external forcing of the climate system, to which extent orbital forcing alone could explain the
change of the dominant periodicity of glacial-interglacial cycle over the geological history remains uncertain. For
instance, the dominant periodicity shifted from ~40 kyr to ~100 kyr over the Mid-Pleistocene transition (MPT) and
the amplitude of the 100 kyr variability increased over the Mid-Brunhes Transition (MBT), but there are no strong
obvious differences in the orbital parameters before and after these transitions (Berger, 1978; Laskar, 2010).
Orbitally-driven internal climatic processes have therefore been proposed to explain these transitions, such as a
meridional shift in the Southern Hemisphere westerlies (Kemp et al., 2010) and changes in Southern Ocean
ventilation and Antarctic bottom water formation (Yin, 2013). Notably, a recent study found that the MBT coincides
with a change of the relative importance of precession and obliquity on high-latitude insolation, while the MPT
aligns with a weakening of both precession and obliquity variations, suggesting a potential orbital origin of these
climate transitions (Berger et al., 2024). Furthermore, orbital variations can drive biome shifts, such as transitions
from shrubland to tropical forest linked to Inter-Tropical Convergence Zone (ITCZ) variability, which may increase
climate sensitivity to orbital forcing—as seen across the Eocene-Oligocene transition (Tardif et al., 2021;
Westerhold et al., 2020). These studies highlight that the relationship between orbital forcing and climate is not
constant in time, possibily due to internal climate feedbacks, boundary conditions and background climate.
The Miocene (~23 to 5.3 Ma) was a warm interval within the long-term Cenozoic cooling trend, marked by stepwise
expansion of Antarctic sea ice and intensified monsoons circulation (e.g., (Steinthorsdottir et al., 2021; Holbourn et
al., 2013; Holbourn et al., 2018; Westerhold et al., 2020). These climatic changes have been linked to orbital
forcing, notably through mechanisms such as Antarctic ice dynamics (Levy et al., 2019; Naish et al., 2009) and
eccentricity-paced variations in the marine carbon cycle associated with an intensified tropical hydrological cycle
(Holbourn et al., 2007; Liu et al., 2024; Tian et al., 2013). Long-term marine records indicate that the sensitivity of
Antarctic ice sheets to obliquity forcing intensified during the Miocene and continued to strengthen through the
Pliocene and Pleistocene (Levy et al., 2019; Van Peer et al., 2024). In addition, spectral analyses of δ$^{18}$O and δ$^{13}$C
proxies indicate dominantat 400 kyr eccentricity-paced variability during the Miocene, with a subsequent transition
toward stronger 100 kyr and 40 kyr cycles [*Holbourn et al.,* 2007; *Tian et al., 2013; Westerhold et al.*, 2020; *Liu et*
*al., 2024*]. Despite these insights, the underlying mechanism remains poorly understood due to a lack of targeted
climate modelling studies.
By conducting climate model simulations, this study examines how the Miocene climate responds to orbital forcing
compared to the pre-industrial (PI) period, providing insight into how climate feedback operates under different





climate states (Steinthorsdottir et al., 2021). In particular, it explores how the absence of NH ice sheets, expanding
Southern Ocean sea ice and strengthening monsoon rainfall shape Miocene orbital-scale climate variability on
orbital scale.
**2 Climate model and simulation setup**
**2.1 FGOALS-g3 climate model and simulation setup**
We use the fully coupled general circulation model FGOALS-g3, part of CMIP6, to perform the simulations. It has
been widely applied to both present-day (Li et al., 2020; Lin et al., 2022; Wang et al., 2020) and paleoclimate
studies from the Miocene to mid-Holocene (Wei et al., 2023; Zheng et al., 2020). Details about the model are
provided in the Supplementary information.
Two baseline experiments were conducted: a pre-industrial (PI) simulation and a Miocene simulation (MI-3x). The
PI simulation was performed with pre-industrial climate forcing. The MI-3x simulation follows the MioMIP2
protocol, and incorporates the reconstructed Miocene boundary conditions, including paleogeography, vegetation,
ice sheet and an atmospheric $CO_2$ concentration that is three times (3x) of the PI level (Burls et al., 2021). The solar
constant, orbital parameters, and aerosol concentration in MI-3x are kept identical to the PI simulation.
To examine climate response to orbital forcing, we conducted sensitivity simulations by modifying orbital
parameters in each baseline experiment. A cold-orbit simulation with minimum NH summer insolation (orbmin),
and a warm-orbit simulation with maximum NH summer insolation(orbmax) were performed for both the PI and the
Miocene (Table S1). Our Miocene simulation focus on the mid-to-late Miocene (11–10 Ma), a period marked by
pronounced δ¹³C excursion and widespread carbon burial associated with the Monterey carbon isotope events
(Anttila et al., 2023; Holbourn et al., 2018; Westerhold et al., 2020). Specifically, we selected 10.777 Ma (insolation
maximum) and 10.767 Ma (insolation minimum) as representative time slices (Fig. S1). This orbital sensitivity
approaches have been widely used in previous Pleistocene studies (Battisti et al., 2014; Bosmans et al., 2018; Dai et
al., 2024).
The NH June insolation contrast between these two cases reaches 130 W/m² at 65 ̊N and 90 W/m² at 20 ̊N (Fig. S1
& S2). This seasonal insolation difference primarily results from the change in the longitude of perihelion: 281 ° and
68 °, corresponding roughly to boreal summer and winter occurring at perihelion, respectively (Fig. S1 & Table S1).
In orbmax, Earth receives more insolation during boreal summer and less during winter, amplifying the annual
insolation cycle by 80 W/m² relative to the baseline. Conversely, orbmin reduces the annual cycle by 60 W/m² (Fig.
1d). Meridionally, orbmax increases annual mean insolation at high-latitudes while slightly reducing it in the tropics
due to its higher obliquity, with the opposite pattern observed in orbmin (Fig. 1c).
Notably, although these configurations are specific to the Miocene, similar orbital patterns also recur throughout the
Pleistocene (Fig. S3). Applying these orbital forcings to the PI and Miocene baseline simulations yields two pairs of
experiments: PIorbmax/PIorbmin for the pre-industrial period and MIorbmax/MIorbmin for the Miocene.



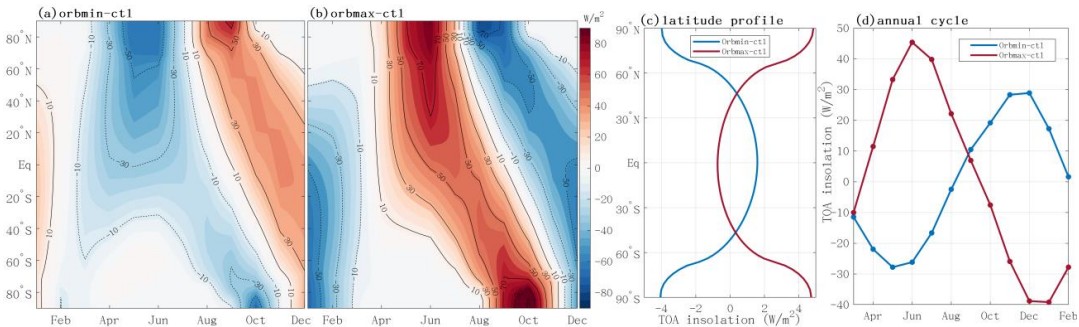


**Figure 1. Orbital-induced insolation changes (W/m²) of the orbmin (a) and orbmax (b) simulations from the baseline**
**simulation, and their latitude profile of annual mean insolation (c) and globally averaged annual insolation cycle (d).**

The PI and MI-3X simulations were each run for 1700 years to reach equilibrium. The orbital simulations were then
branched from the year 1601[th] of the PI and Miocene run and integrated for additional 300 and 400 years,
respectively. This approach ensured that the global mean Top-of-Atmosphere (TOA) radiation imbalance within
±0.34 W/m² (Table. S1) over the final 100 years. Monthly output from these equilibrated simulations were used for
subsequent analysis. The PI simulation reasonably captures seasonal temperature variations in both magnitude and
spatial pattern, closely matching the CMIP5 multi-model mean and ERA5 data despite a minor cold bias in Arctic
Eurasia due to sea ice overestimation (see SI for more details).
**2.2 Diagnostic analysis**
We conducted a one-dimensional Energy Balance Model ananlysis (EBM)(Heinemann et al., 2009; Wei et al., 2023)
to quantify how changes in radiative components contribute to temperature response to orbital forcing. The EBM
balances net incoming shortwave radiation with outgoing longwave radiation and meridional heat transport, using
radiative fluxes from the GCM as input. Temperature differences between simulations are decomposed into
contributions from change in greenhouse gas of water vapor, surface albedo, and heat transport, and cloud. The
cloud effects can be further decomposed into shortwave and longwave components.
The EBM components align well with the GCM simulated results, effectively capturing the zonal-mean features of
temperature response, with deviations of 0.1-0.9 ℃ (Fig. S4 ). Slight underestimations occur in the NH subtropics,
polar regions, and high-latitude Southern Ocean, while an overestimation appears around 70-80 ̊N latitudinal (Fig.
S4). These slight discrepancies are similar in magnitude to those reported in the previous studies and mainly arise
from seasonal and zonal averaging of nonlinear processes (Lunt et al., 2012).



## 3 Results and Discussion

### 3.1 Weaker seasonality of temperature response during the Miocene

Orbital forcing modulates seasonal temperature variations. Reduced boreal summer insolation weakens the seasonal cycle, from 3.2 °C in MI-3x to 1.3 °C in MIorbmin, and from 3.7°C in PI to 1.6 °C in PIorbmin. Conversely, increased boreal summer insolation intensifies seasonality, reaching 5.4 °C in MIorbmax and 6.4 °C in PIorbmax (Fig. 2). Consequently, seasonal GMAT variations rise by more than 2 °C in the orbmax simulations and decline by a similar magnitude in the orbmin simulations relative to their respective baselines. These changes in JJA temperature differences exceed 2.5 °C between orbital simulations (Fig. 2)—a shift comparable to the ~3 °C global cooling during the late Miocene (Westerhold et al., 2020), underscoring the role of orbital forcing in climate variability.

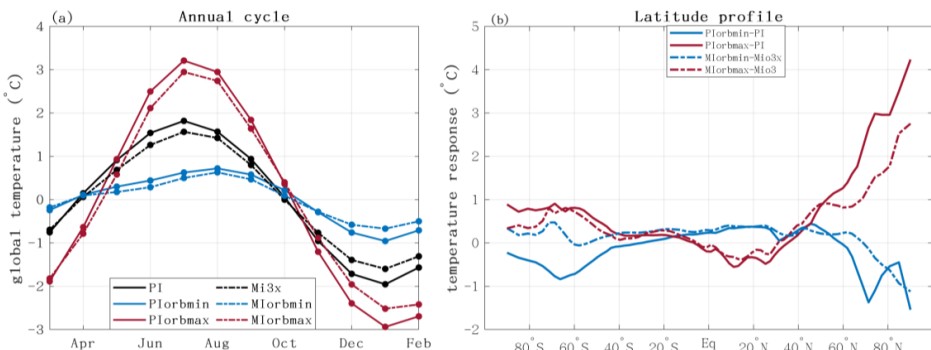

**Figure 2. Annual cycle of temperature anomalies (from their annual mean) (a), and their latitude profile of temperature response to orbital forcing (b).**

Compared to the PI context, the MI-3x simulation exhibits weaker seasonality and a dampened orbital response (Fig. 2a). The annual cycle is smaller in MI-3x than in PI, reflecting reduced July warming and January cooling. The GMAT response to orbital forcing is diminished by approximately 0.1 °C in both Miorbmin and MIorbmax simulations, leading to ~10 % weaker seasonal amplitude changes. This diminished Miocene temperature response is also evident in the latitudinal profile, showing differences of up to 1 °C at high latitudes (Fig. 2b).

This reduced Miocene seasonality is consistent with proxy-based evidence indicating lower seasonality during the warming Miocene in Europe (Harzhauser et al., 2011), the Mediterranean (Utescher et al., 2009), and N America (Reichgelt et al., 2023). Variation in Miocene's seasonal response to identical orbital forcing can alter the relationship between growing-season and annual mean temperatures, potentially biasing proxy-based climate reconstructions. This highlight the importance of applying seasonality adjustments that account for different paleoclimate contexts, rather than relying solely on modern analogs, when addressing well-documented seasonal biases in proxies (Bova et al., 2021; Marsicek et al., 2018; Laepple and Lohmann, 2009; Laepple et al., 2022).



### 3.2 Spatially varied Miocene temperature responses


The orbmax and orbmin simulations show overall anti-phased annual mean temperature responses (Fig. 3).
Compared to the MI-3x and PI baselines, orbmax simulations show a dipole pattern, with polar warming but cooling
in the tropics and subtropics of both hemispheres. Conversely, orbmin simulations show an opposing dipole: cooling
at high-latitudes and warming in the tropics, extending up to ~60 °N and 40 °S. Similar high-low latitude contrasts
have been reported in simulations of interglacials characterized by high obliquity and precession, such as Mid-
Holocene (Brierley et al., 2020; Dai et al., 2024) and other interglacials [Yin and Berger, 2012; Herold et al 2012].
These patterns are mainly related to the change in obliquity and precession, and are further amplified by feedback
including high-latitude albedo changes and shifts in the tropical hydrological cycle (Fig. S5 & S6).

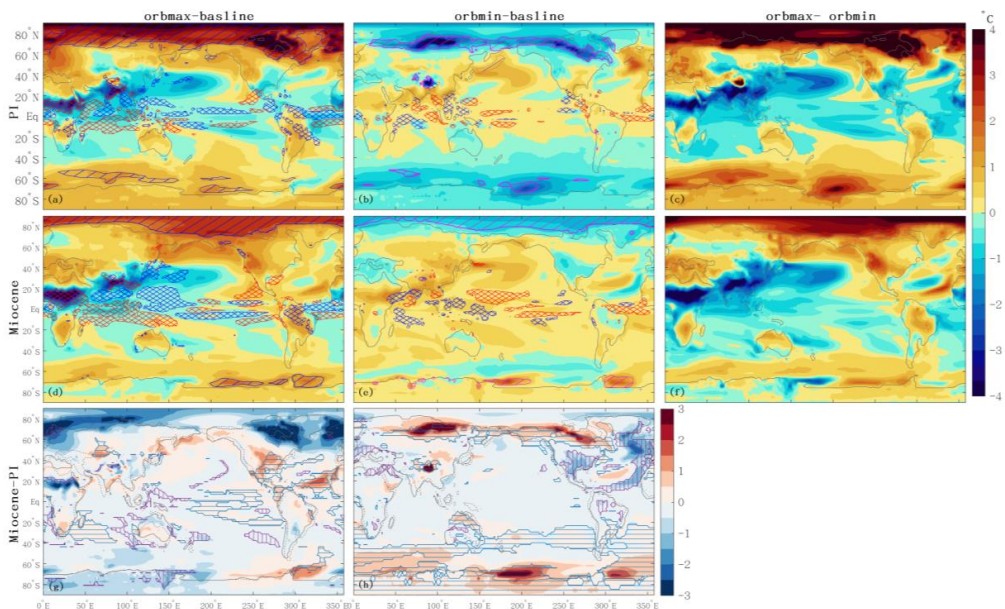


**Figure 3. Annual mean air temperature response to orbital forcing, with anomalies relative to baseline**
**simulations displayed in the top two panels. Crosses mark areas where precipitation increased (red) or**
**decreased (blue) by more than 0.6 mm/day. Hatching indicates regions where albedo increase (magenta) or**
**decrease (blue) by over 5%. The last column summarized orbmax-orbmin differences. The bottom panel**
**highlights differences between PI and Miocene response; blue horizontal and purple vertical hatching**
**marking regions where the sign of anomalies is reversed —shifting from negative in PI to positive in Miocene,**
**and vice versa.**

### 3.2.1 Reduced High-latitude Orbital Response in the Miocene


Compared to PI, the Miocene orbital response is notably weaker at high northern latitudes (Fig. 3). Under the PI
context, the strongest warming in PIorbmax (~4.8 °C) occurs over northeast Canada and the Labrador Sea, whereas
the MIorbmax warming is less than half as strong (Fig. 3b). Similarly, cooling in the PIorbmin simulation reaches



4.4 ℃ over Western Siberia, but only 1 ℃ in MIorbmin (Fig. 3a, 3c). The strongest Miocene orbital response —
2.8℃ over the Chukchi Sea—is still weaker than in PI.
The EBA results show that much of the weaker Miocene temperature responses in NH high-latitudes can be
attributed to smaller changes in surface albedo (Fig. 4). In the Miocene, the albedo contribution is roughly half of
that in the PI. For example, albedo-driven warming reaches  6 ℃ in PIorbmax but only 3 ℃ in MIorbmax. Similarly,
MIorbmin shows poleward shift and weaker albedo-driven cooling than PIorbmin. This reduced albedo feedback
dampens the Miocene temperature responses to orbital forcing, with temperature changes closely matching the
spatial pattern of albedo variations (Fig. 3). Further analysis  (Fig. S6) reveals that strong albedo changes under the
PI context coincides with ice sheets and sea ice, where ice–albedo feedbacks enhance the climate's response to
orbital forcing. By contrast, the warmer Miocene climate, characterized by widespread vegetation, limited sea ice,
and lower surface albedo, is less sensitive to orbital forcing.

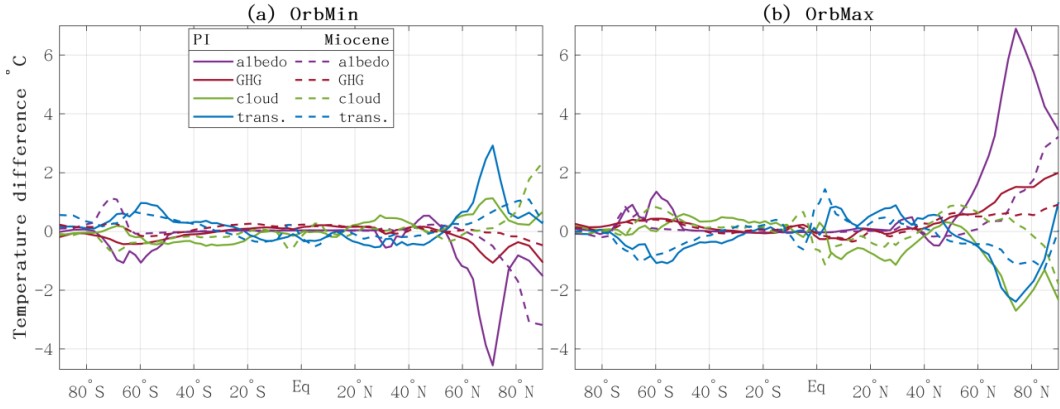


**Figure 4. Zonal mean surface temperature responses (to orbital insolation) of various components from energy balance analysis. Total response is decomposed into contributions from the surface albedo (albedo), water vapor's greenhouse (GHG), heat transport (trans), and cloud  effects (cloud).**

The above strong albedo response is further reinforced by water vapor's greenhouse effect but is counteracted by
cloud cover. Water vapor contributions follow albedo patterns, reflecting their dependence on surface energy
availability, whereas clouds exert an opposing influence. Further decomposition into shortwave and longwave
components reveal that shortwave radiation dominates, indicating a generally weaker net negative feedback in the
Miocene (Fig. S4).
These results are consistent with previous studies suggesting weaker climate sensitivity during warm periods [*De*
*Vleeschouwer et al.*, 2017; *Levy et al.*, 2019; *Naish et al.*, 2009](Reichgelt et al., 2023). For example, proxy
reconstruction indicates muted Miocene climate variability in eastern North America compared to the modern era
(Reichgelt et al., 2023). The larger and more regular temperature variations in the PI simulation suggest an enhanced
sensitivity to orbital forcing, consistent with the development of pronounced NH glacial-interglacial cycles. In



contrast, the Miocene's dampened response implies that climate variability was likely less periodic or characterized
by a more prolonged cycle under warmer climate conditions.

**3.2.2 Enhanced tropical North Africa cooling in the MIorbmax simulation**

An exception to the weaker Miocene response is the enhanced tropical North Africa cooling in the MIorbmax
simulation (Fig. 3d), which shows a 4.4 °C decrease—greater than the 3.8 °C in PIorbmax—and extends farther
north. Seasonal decomposition indicates that this cooling persists even during summer, despite increased insolation
(Fig. S1 & S7). It coincides with intensified precipitation, pointing to a dominant role of insolation-driven
hydrological change in modulating temperature.
The EBM results show that the stronger cooling in the MIorbmax simulation arises from larger water vapor and
cloud changes (-0.34 and -1.12°C in Miocene vs. 0.28 and 0.94°C in PI). These enhanced Miocene cooling effects
align with increased precipitation. Additional analysis of water flux divergence suggests that more moisture from the
Tethys Sea during the Miocene feeds this region's precipitation (Fig. S8). This highlights that a wider Tethys Sea
provides moisture source, while a warmer climate accelerates the hydrological cycle (Fig. S8) (Sarr et al., 2022;
Huntington, 2006). These findings align with proxy evidence for intensified hydrological cycle and increased
precipitation under warm climate, such as the "green Sahara" during the mid-Holocene (Hoelzmann et al., 2001;
Kutzbach and Liu, 1997; Liu et al., 2024), supporting the conclusion that MIorbmax cooling is driven by
hydrological intensification. Similar deeptime sensitivity to orbital forcing has been noted in previous studies, which
found substantial precipitation responses during the early Cenozoic, comparable to monsoon signals (Zhang et al.,

213    2024).

**3.2.3 Disrupted Southern Ocean orbital signal in the Miocene**

In the Southern Ocean, the Miocene response to orbital forcing deviates from the expected anti-phase pattern
observed in PI that roughly follows insolation change. Specifically, unexpected warming occurs over the Ross Sea
and Weddell Sea in response to overall reduced local annual mean insolation in the MIorbmin simulation, contrasted
with the cooling observed in PIorbmin.
This warming is particularly evident in the austral winter (Fig. S10), disrupting the expected anti-phase signal and
generating an out-of-phase pattern. According to EBM analysis, the reversed temperature response in MIorbmin
(1.1°C warming at 71°S instead of cooling) is mainly attributed to albedo and water vapor effects (Fig. 4). During
the Miocene, the maximum sea ice edge extended polarward around 70°S, where significant insolation changes (Fig.
S9) facilitated increased austral winter insolation to trigger positive ice-albedo feedback—reducing sea ice and
enhanced warming (Fig. 3 & S10) ─ and generating more water vapor. In contrast, PI sea ice extended into lower
latitudes where insolation changes were smaller, reducing sensitivity to the seasonal change of orbital forcing (Fig.
S10).
These results support geological evidence that ocean-atmosphere-ice sheet interactions amplified Antarctic ice
sheets sensitivity to orbital forcing during their Miocene expansion [*De Vleeschouwer et al.*, 2017; *Levy et al.*, 2019;



*Naish et al.*, 2009]. Sediment records further indicate a stronger and more stable climate response with growth of ice
sheet and sea ice (De Vleeschouwer et al., 2017; Levy et al., 2019; Naish et al., 2009; Reichgelt et al., 2023),
compared to warmer periods, like the Early Eocene, when intensified carbon-climate coupling played a dominant
role (Setty et al., 2023). Overall, these findings underscore how Southern Ocean sea ice feedbacks are strongly
modulated by the background climate state (Bloch‑Johnson et al., 2021).

**3.3 Irregulate Miocene responses to orbital forcing and reduced internal climate variability**

Beyond differences in magnitude, the spatial extent of warming and cooling response differs notably between the PI
and Miocene. In the PI, high-latitude temperature responses are strictly anti-phased between two insolation cases. In
contrast, the Miocene simulations show warming in both MIorbmin and MIorbmax over Siberia and Alaska,
resulting in distinct regional responses across 60-70 °N, spanning Eurasia, Alaska and North America continent (Fig.
3e, blue line). Similarly, the Weddell Sea and Ross Sea in the Southern Ocean show overall warming in both orbital
simulations, deviating from the expected anti-phased changes observed in the PI (Fig. 3 & S11). These deviations
suggest a less predictable Miocene climate system, likely reflecting the dominance of longer periodicities, such as
the 400-ka cycle, rather than the 40 ka and 10 ka cycle of the Pleistocene, as suggested by proxy records [Holbourn
et al., 2007; Tian et al., 2013; Westerhold et al., 2020; Liu et al., 2024]. This also implies that simply comparing
orbmax minus orbmin, without examining spatial patterns in detail, risks missing nonlinear responses—particularly
under warmer climate conditions.
In addition to seasonal changes, internal temperature variability is crucial for assessing climate stability (Harzhauser
et al., 2011). To investigate this, we examine deseasonalised variability (expressed as the standard deviation) across
above key high-latitude regions where major glacial dynamics occur. Results show that mid-latitude Eurasia and
North America exhibit higher internal temperature variability in the PI simulation, which is further amplified under
both PIorbmin and PIorbmax, reflecting such as ice-albedo interactions and land-atmosphere coupling. By contrast,
the Miocene showed lower variability in those regions, further reduced in the MIorbmin simulation (Fig. 5),
suggesting a more stable high-latitude climate with dampened feedbacks under warmer background conditions.
Reduced Eastern Pacific variability in the Miocene likely reflects enhanced inter-basin exchange through an open
Panama Seaway, which buffers regional responses. Stronger and more regular temperature variability in the PI
simulations indicates greater orbital sensitivity and support the development of pronounced NH glacial-interglacial
cycles, whereas the dampened and irregular Miocene response suggests weaker or less periodic cycles under warm
conditions.



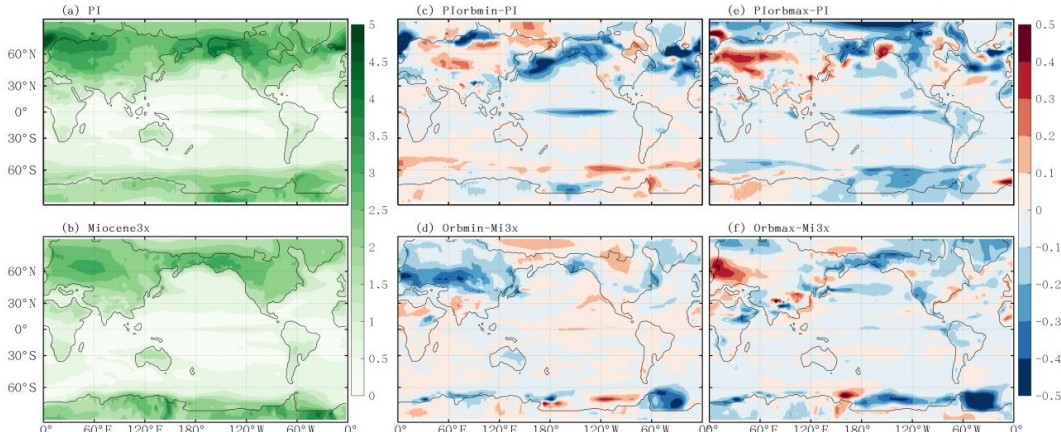


**Figure 5. Standard deviation of deseasonalized temperature, and their responses to orbital forcing.**

**4 Conclusions and Implications**

Orbital insolation drives glacial-interglacial cycles through complex feedbacks involving continental ice sheets. While most studies have focused on the Quaternary—when large continental ice sheets presented in both hemispheres—the climate response to orbital forcing during warm periods, such as during the Miocene, remains less well understood. This study addresses that gap by conducting and comparing two sets of orbital sensitivity simulations under pre-industrial (PI) and mid-Miocene conditions, each forced with minimum (orbmin) and maximum (orbmax) boreal summer insolation.

The simulations reveal an overall anti-phased temperature response between the two orbital scenarios: Orbmax leads to high-latitude warming and tropical cooling, whereas orbmin produces the opposite pattern. However, compared to PI, the Miocene exhibits a weaker global temperature response (~1℃ smaller) and less stable anti-phased behavior among orbmax and orbmin. Key differences include: (1) Weaker temperature response over Northern Hemisphere continents, caused by reduced surface albedo feedbacks primarily due to different vegetation and ice cover, along with contributions from water vapor greenhouse effect and cloud cover; (2) Stronger cooling (4.4℃) over North Tropical Africa in the MIorbmax simulation, compared to 3.8℃ in PIorbmax, driven by enhanced hydrological cycle changes due to more moisture from a wider Tethys Sea under the warmer background climate; (3) Out-of-phase temperature response in the Southern Ocean in the MIorbmin simulation, driven by increased austral winter insolation, sea ice reduction, and subsequent positive ice-albedo feedbacks.

These distinct responses have two major climate implications: First, the weaker Miocene seasonal response alters the difference between growing-season and annual mean temperatures. This mismatch may lead to overestimates of annual mean temperature reconstruction based on some proxy records, highlighting the need for context-specific seasonality corrections rather than reliance on modern analogs (Bova et al., 2021; Marsicek et al., 2018; Laepple and Lohmann, 2009; Laepple et al., 2022). Second, the larger and more regular temperature variability in the PI context



indicates stronger sensitivity to orbital forcing and supports the presence of well-developed NH glacial–interglacial
cycles. In contrast, the Miocene's subdued and irregular climate variability suggests weaker and less periodic orbital
pacing under warmer conditions, consistent with proxy evidence [*Holbourn et al.*, 2013; *Holbourn et al.*, 2018;
*Steinthorsdottir et al.*, 2021; *Westerhold et al.*, 2020].

**Acknowledgments**
This study was supported by the National Key R&D program of China (2023YFF0803902) and (2023YFF0803904).
We appreciate the technical support of the National Large Scientific and Technological Infrastructure, *Earth System*
*Numerical Simulation Facility* (https://cstr.cn/31134.02.EL).

**Open Research**
Model output data from this study are available at Zhang (2025).

**Author contributions**
Conceptualization & Study Design: YZ;
Methodology & Simulations: YZ, JW, with support from WZ;
Formal Analysis & Investigation: YZ, with guidance from YQ, A. de B., ZS, LZ;
Data Curation: ZL, ND;
Writing – Original Draft: YZ;
Writing – Review & Editing: All authors.

**Competing interests**
Some authors are members of the editorial board of journal Climate of the Past.

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
