# Peer review of "Weakened and Irregular Miocene Climate Response to Orbital"

_EGUsphere, 2025_

## Author Comment (AC2)

**RC2**: 'Comment on egusphere-2025-4485', Anonymous Referee #2

**Review:** Zhang et al., Weakened and Irregular Miocene Climate Response to Orbital Forcing compared to the modern day

**Summary**

This manuscript explored the impact of orbital forcing during the Miocene. The authors provided a comparison between a preindustrial and a high CO2 middle Miocene simulation and various sensitivity experiments with orbital min and max configuration. The authors suggest a weaker seasonality response to orbital configuration primarily due to the weak response of surface albedo feedback. Although the results presented are interesting, the current version of the manuscript presents more questions than answers. This is mostly due to insufficient analysis being presented.

We thank the reviewer for the insightful comment. Please find detailed response below.

**Major**

From a first-principle standpoint, why does your middle Miocene run have a weaker seasonality? Is it CO2, paleogeography, or ice-sheet configuration? The author raised all of these things in the introduction, but doesn't really provide any answers.

Thank you for raising this key question. Although a full attribution is beyond the scope of this study, we added result from an additional simulation (MCO-1x: same boundary conditions but lower pCO2) shown in Fig. S4. The MI-1x seasonality (~3.5 °C) lies between PI (3.7 °C) and MCO (3.2 °C), indicating that both elevated $CO_2$ and Miocene boundary conditions contribute to the reduced seasonality, with the latter exerting a slightly larger influence. We added a sentence in Section 3.1 to clarify this.

It seems apparent in Figure 2 that your baseline MCO run shows an overall weaker seasonality, so in turn, your other sensitivity experiments also have a similar response to orbital changes. This leads to the question, is it because your PI run have lower CO2 that is leading to a stronger seasonality? Is it a general statement that warm climate intervals have weak seasonality or is it unique to the MCO?

Thanks for this interesting point. Fully disentangling the role of background $CO_2$ and broader warm-climate mechanisms would require a larger ensemble spanning multiple $CO_2$ levels and additional warm intervals (e.g., MioMIP + PlioMIP). In this study, our aim is more limited: we show that under identical insolation anomalies, the MCO simulation exhibits a weaker temperature response than PI. Whether this reduced orbital sensitivity reflects a general feature of warm climates or is specific to the MCO cannot yet be determined. We now clarify this explicitly in the revised manuscript: "Because comparable analyses are not yet available for other warm climate intervals, it remains uncertain whether the reduced orbital response identified here is specific to the MCO or reflect a more general feature of warm climate states. This question requires further investigation.".

Although it is interesting to see a weaker surface albedo response in the MCO simulations, it should be noted that this feedback is inherently linked to the prescribed vegetation and land ice. It is really only sea-ice and potentially cloud feedback that's

responding to the orbital changes. The authors should show which parameter is causing the large albedo change. I assume from Figure S6 that sea-ice in the PI is responding much more readily, where your MCO runs most likely do not have any sea ice.

Thank you for this helpful comment. Because surface albedo in the model is computed diagnostically (reflected/incoming shortwave), land ice, vegetation and sea ice cannot be perfectly separated.

Regarding whether MCO has sea ice, the MCO simulations do retain seasonally varying sea ice despite with greatly reduced perennial ice, as shown in Fig. S8. Thus, this seasonal ice still responds to orbital forcing, but its variability—especially in NH—is much weaker than in the PI, where extensive sea ice allows a much stronger albedo feedback. This partly explains the stronger PI response.

In the Southern Ocean, limited but sensitive Miocene winter sea ice can still generate local positive ice–albedo feedbacks (e.g., in MCOorbmin), so the sea-ice contribution is region-dependent.

We have clarified this in Section 3.2.1 and 3.2.3.

It would be useful to see how the SST, deep ocean, and various MOC respond to the orbital changes. I suspect this could be one of the reasons why you have such a weak climate response. For example, the PI run would most likely have a strong AMOC and could be easily impacted by orbital changes, while your MCO 3x simulation does not. Also, the authors primarily use ocean proxy evidence to indicate a weaker orbital response; its only appropriate the authors should supply some type of ocean analysis.

Thank you for raising this important point. We agree that examining the oceanic response—particularly SST, deep-ocean temperatures, and the overturning circulations—would provide valuable context for interpreting the climatic sensitivity to orbital forcing. A full analysis of the ocean circulation is substantial and is being prepared in a companion paper focused specifically on Miocene ocean–atmosphere dynamics.

To address the reviewer's concern here, we showed Atlantic Meridional Overturning Circulation (AMOC) response in our simulations (see figure below). Orbital forcing induces only modest AMOC anomalies in both the PI and Miocene experiments. Importantly, the Miocene AMOC is already weaker and shallower, lacking a strong deep North Atlantic branch. This reduced overturning diminishes the system's ability to amplify orbital forcing, consistent with the weak global temperature response.

We now highlight this in the discussion and state that SST and deep-ocean analyses will appear in the forthcoming study.

[Figure]

Fig. 1 MOC streamfuction in the simulations.

The author should modify the use of the general term "Miocene" to either middle Miocene or MCO since the boundary condition utilized in the experiments does not represent paleogeography, vegetation, ice sheet and etc changes in the late Miocene.

We agree and now refer to the simulations consistently as "MCO" to reflect that the boundary conditions correspond to the Miocene Climatic Optimum rather than the entire Miocene epoch.

**Minor**

The title is a bit misleading since regardless of the orbital changes with or without your MCO runs have a weak seasonality; nothing about it is irregular.

we agree that even if we do not consider the change of orbital forcing, the MCO base run already has a weaker seasonality than PI, but what we stress here is a weaker climate response to orbital forcing during the Miocene (which might be partly linked with its weaker seasonality?). Regarding the word "irregular", we indicate the response in some region is stronger and in some region is reversed and have replaced it with "diverse".

Line 57 extra parenthesis

This has been fixed.

Line 66 vague sentence. Mechanism for what? Also, plenty of examples of Miocene modeling targeting specific mechanisms including orbital forcing. A generic statement is a bit disingenuous.

We mean there is no modelling work to specifically insolate orbital-driven variation for Miocene. This has been clarified as "Although geological archives provide evidence for persistent orbital pacing during the Miocene, the mechanisms linking these

variations to climate response—particularly in warm climates lacking large Northern Hemisphere ice sheets—remain poorly constrained. In particular, there is a scarcity of climate modelling studies that isolate orbital effects under realistic Miocene boundary conditions."

Lines 106-112 TOA imbalance of .34 suggest not fully equilibrated I would suggest modifying "reach equilibrium" to quasi-equilibrium.

True, this sentence has been accordingly modified.

Line 190 citation needs to be fixed.

This has been done.

Line 269 I'm not sure what you mean by "less stable anti-phased behavior" ? Please provide a timeseries that shows fluctuation or instability in mean climate. From your results overall weaker seasonality would suggest much more stable climate.

Aplologies for confusion. We mean the orbmax and orbmin simulations have opposite response in PI. But in MCO, they do not appear as expected. We have clarified this as "Both climates exhibit broadly anti-phased temperature response between maximum and minimum boreal summer insolation, but the Miocene response is ~1°C weaker, spatially less coherent, and shows greater regional diversity".